# The Production of Pyruvate in Biological Technology: A Critical Review

**DOI:** 10.3390/microorganisms10122454

**Published:** 2022-12-12

**Authors:** Wei Yuan, Yongbao Du, Kechen Yu, Shiyi Xu, Mengzhu Liu, Songmao Wang, Yuanyuan Yang, Yinjun Zhang, Jie Sun

**Affiliations:** Key Laboratory of Bioorganic Synthesis of Zhejiang Province, College of Biotechnology and Bioengineering, Zhejiang University of Technology, Hangzhou 310014, China

**Keywords:** pyruvic acid, yeast, *E. coli*, biochemical pathways, microbes

## Abstract

Pyruvic acid has numerous applications in the food, chemical, and pharmaceutical industries. The high costs of chemical synthesis have prevented the extensive use of pyruvate for many applications. Metabolic engineering and traditional strategies for mutation and selection have been applied to microorganisms to enhance their ability to produce pyruvate. In the past decades, different microbial strains were generated to enhance their pyruvate production capability. In addition to the development of genetic engineering and metabolic engineering in recent years, the metabolic transformation of wild-type yeast, *E. coli*, and so on to produce high-yielding pyruvate strains has become a hot spot. The strategy and the understanding of the central metabolism directly related to pyruvate production could provide valuable information for improvements in fermentation products. One of the goals of this review was to collect information regarding metabolically engineered strains and the microbial fermentation processes used to produce pyruvate in high yield and productivity.

## 1. Introduction

Pyruvic acid or Pyruvate (PA) is the final product of the glycolysis pathway and is an important intermediate molecule in protein, fat and carbohydrate metabolism [1]. PA is a three-carbon molecule that functions as a critical step in cell metabolism in aerobic and anaerobic conditions [2]. The two PA molecules produced from glycolysis can be converted into various products, like carbohydrates (through gluconeogenesis), alanine, or ethanol (through fermentation), and fatty acids and energy (through Krebs cycle) [1]. Inside the cells, various metabolic pathways are connected by PA [1]. PA is a transparent, water-miscible liquid having an acetic acid-like aroma under normal conditions. PA and its derivatives are widely employed in the pharmaceutical, cosmetic, food, and other important industries [3,4]. PA is also found in sports nutrition supplements, which help athletes to improve their physical condition and maintain body mass control [5]. PA calcium salts were demonstrated to speed up fatty acid metabolism, while also helping to reduce serum cholesterol levels [6]. PA also possesses antioxidant effects and could be used as a nutraceutical to treat diabetes II [7,8].

At the industrial level, pyruvate is mainly used for the production of L-tyrosine, L-tryptophan, N-acetylneuraminic acid (sialic acid), 3-Dihydroxyphenyl (DOPA), levodopa drug class, etc. [9,10]. Current research shows that the microorganisms that ferment to produce pyruvate are mainly *E. coli* and yeast. In *E. coli*, lactate dehydrogenase (ldhA), which is responsible for converting pyruvate into lactic acid, is inactivated, and the highest pyruvate production is 110.0 g/L with a mass yield of 0.87 g/g. In yeast, Pyruvate decarboxylase-negative *saccharomyces cerevisiae*, through directed evolution, obtained a high-yield pyruvate strain that was C2-independent and glucose-tolerant, with a maximum yield of 135 g/L [11]. The pyruvate/glucose conversion rate was 0.54 g/g. Using 70 g/L NaCl as the selection criterion, the NaCl-resistant mutant *Torulopsis glabrataRS23* was screened out through continuous culture with pH control. The yield of pyruvate reached 94.3 g/L, and the mass yield was 0.635 g/g. Most of the above-mentioned strains with high pyruvate production use glucose as a substrate, and achieve high production of pyruvate by reducing the consumption of pyruvate. However, pyruvate production mainly uses a chemical synthesis method, which increases its cost and greatly limits its application. In past decades, different microbial strains were generated to enhance their pyruvate production capability. This review summarizes the methods and tools for the production of pyruvate, by the engineering of different microorganisms, conducted in recent years.

## 2. General Methods of Pyruvic Acid Production

### 2.1. Chemical Method

At present, the synthesis methods of pyruvic acid are mainly chemical methods, such as the tartaric acid method [3], the lactic acid oxidation method and other methods. Among the methods, the tartaric acid method mixes and heats tartaric acid with potassium pyrosulfate, and, finally, the tartaric acid is dehydrated and decarboxylated to form pyruvic acid. According to the latest market situation in 2022, the price of tartaric acid alone reaches about $3000/t. The most important thing is that this method produces serious environmental pollution. Another method uses lactic acid as a raw material and causes it to react with oxygen under the action of a catalyst to produce pyruvic acid. This is a chemical synthesis method, used in many current research works. As the C–C bond is easy to break, producing acetaldehyde and CO_2_, during the catalytic oxidation of lactic acid, the research is mainly carried out in the comparison and selection of catalysts. With the Pb/Pd/C system as the catalyst [12], the yield of pyruvic acid is relatively ideal. The conditions required for the reaction are mild, green and environment-friendly, and oxygen can be replaced with air in industrial production, which has been gradually adopted by more and more factories in the world.

### 2.2. Biotechnological Methods

Compared to the chemical approach for pyruvate production, biotechnological methods, including enzymatic, resting cell and fermentative processes, offer a promising alternative for cost-efficient process development. The basic process is shown in Figure 1.

### 2.3. Enzymatic Processes

This process involves a single step carried out by either purified or raw enzymes (or, sometimes, whole cells). *Acetobacter* sp. has been used to produce about 20 g/L pyruvate by oxidizing D-lactate at a high conversion rate. However, this process is difficult to commercialize, as D-lactate costs more than L-lactate. Another process for pyruvate production involves oxidation of L-lactate to pyruvate by using glycolate oxidase (obtained from *Hansenula polymorpha*. EC:1.1.3.15) [13].

### 2.4. Resting Cell Processes

Resting cell processes involve a series of enzymatic steps in non-growing microbial cells provided with a substrate, such as glucose. *Acinetobacter* sp. and *Debaryomyces cerevisiae* have been used to produce pyruvate by the resting cell method. This method consumes less time than the fermentation one; however, it involves different steps like cell cultivation, separation from the medium and washing. However, few resting cell methods are performed without separating cells from the medium. By controlling pH or nitrogen source, cell growth can be inhibited. The major issues associated with the extended resting cell processes, includes strain stability and contamination.

### 2.5. Fermentation Processes

Direct fermentation processes have also been used to produce pyruvate. Many recombinant microorganisms, including yeast such as *Y. lipolytica* and *E. coli*, have been developed to accumulate pyruvate from different carbon sources. For example, as an oil producing yeast, *Y. lipolytica* has a relatively complete metabolic network in its body, so it can use gene editing technology to flow more metabolic flux into the production of pyruvate. These strains are discussed in detail in the section below.

## 3. Biochemical Pathways Involved in Pyruvate

Pyruvate is a key metabolite synthesized via glycolysis, and it occupies an important position in the central pathways [14]. The biochemical pathways directly involved in the formation and assimilation of pyruvate are shown in Figure 2. During growth of *E. coli* on glucose, pyruvate is generated mainly from Phosphoenolpyruvate (PEP) by pyruvate kinase (EC:2.7.1.40) and by the phosphotransferase system (PTS) of glucose uptake [15]. Aerobically, pyruvate is converted into acetyl CoA by pyruvate dehydrogenase (PDH.EC:1.2.1.51); however anaerobically, this conversion is carried out by formate-lyase (PFL.EC:1.97.1.4). Pyruvate can also be converted into acetate, malate and PEP by pyruvate oxidase (POX.EC:1.2.3.3), malic enzyme and PEP synthase (PPS.EC:2.7.9.2), respectively. In addition to these directly related pathways, many adjacent pathways, and regulatory proteins, are likely to impact the accumulation of pyruvate, such as PEP carboxylase (PPC.EC:4.1.1.31), Phosphotransacetylase (EC:2.3.1.8), Acetyl-CoA synthetase (EC:6.2.1.1), acetate kinase (EC:2.7.2.1), and the glyoxylate shunt (Figure 1).

### 3.1. Biochemical Pathways Directly Impacting Pyruvate

The phosphotransferase system (PTS) is the major pathway in which a large number of sugars are phosphorylated and transported into the cell [16]. This system governs regulation of other metabolic pathways as well [16]. Pyruvate is the product of the reaction. The overall reaction of PTS can be described by the following equation for numerous microorganisms and carbohydrates:
PEP + Carbohydrate (ext) → PTS − Pyruvate + Carbohydrate-P (int)

The PTS is composed of two soluble proteins; enzyme I and histidine phosphate carrier protein (HPr), and sugar specific protein(s), enzyme II [17]. Enzyme I and HPr are responsible for the phosphorylation of all carbohydrates in PTS. Therefore, it is called the general PTS protein, while enzymes II are carbohydrate-specific enzymes [16]. There is a phospho-transfer cascade in PTS. Enzyme I is a ~64 kDa protein, which can be phosphorylated on a histidine residue (His189) by PEP in the presence of Mg^2+^ [18]. HPr (~9 kDa) is able to accept a phosphoryl group from phosphorylated enzyme I on its His15. Then, the phosphorylated HPr is capable of phosphorylating sugar-specific enzyme II. In the case of glucose, a soluble, cytoplasmic protein, phosphate carrier protein (IIAGlc) accepts the phosphoryl group on His90. Then, a membrane-associated protein, enzyme IIB-IIC, transfers the phosphoryl group from phosphorylated IIAGlc to glucose and transports it into cells [18]. There are several proposed feedbacks associated with regulation mechanisms for PTS, including the membrane potential, energy-dependent efflux of substrates, competition for HPr-P, and regulation by intracellular phosphor-compounds [16].

After glucose is converted into glucose-6-phosphate and transported into the cell, it enters glycolysis, the stepwise degradation of glucose into pyruvate. There are 9 reactions and 9 enzymes involved in glycolysis, starting from glucose-6-phosphate. The final enzyme involved in glycolysis is pyruvate kinase (PYK). Pyruvate kinase catalyzes the conversion of PEP and ADP to pyruvate and ATP [19]. *E. coli* expresses two PYKs; namely PYKF (encoded by *pykF*) and PYKA (encoded by *pykA*) [20]. The two isoforms actively play a role in pyruvate biosynthesis, when glucose is the sole carbon source [20]. PYKF contributes more to pyruvate biosynthesis, compared to PYKA. The catalytic activity of PYK is regulated by PEP substrate [19]. PYKF is activated by fructose-1,6-bisphosphate (FBP) and inhibited by ATP, while PYKA is activated by monophosphorylated sugars and AMP [19].

It was found that pyruvate production increased as a result of heat treatment in *M. robertsii* through pyruvate kinase upregulation. Similarly, downregulation of genes, like *MAA_02871* and *MAA_08787*, enhanced pyruvate production [21]. *MAA_02871* encodes a pyruvate transporter, while *MAA_08787* encodes the β-subunit of the pyruvate dehydrogenase E1 component. a new *E. coli* mutant. Mutations in phosphoenolpyruvate carboxylase (PPC), pyruvate dehydrogenase (PYD), and alcohol dehydrogenase (ADH) enhance pyruvate production [22]. Furthermore, it was found that thiamine or lipoic acid auxotroph can accumulate pyruvate [23]. The auxotroph grown in a lipoic acid or thiamine-limiting medium reduces the specific activity of the pyruvate dehydrogenase complex (PDHC), and, thereby, enhances pyruvate accumulation.

#### 3.1.1. Pyruvate Dehydrogenase (PDH)

The conversion of pyruvate to acetyl-CoA is the connecting link between glycolysis and the TCA cycle. This reaction is catalyzed by pyruvate dehydrogenase complex, a sixty submit complex involving three enzymes: pyruvate dehydrogenase, dihydrolipoyl transacetylase and dihydrolipoyl dehydrogenase. These enzymes are associated noncovalently. The first step of the reaction is performed by pyruvate dehydrogenase (E1): pyruvate releases CO_2_ and binds to thiamine pyrophosphate (TPP), forming hydroxyethyl TPP (HETPP). The pyruvate dehydrogenase (E1) is a hetero tetramer (α2β2) with a total molecular weight of 154 kDa, and is encoded by the gene *aceE*. The second step is catalyzed by dihydrolipoyl transacetylase (E2), which involves acetyl group transfer to CoA. E2 is encoded by the *aceF* gene. Finally, the reduced dihydrolipoamide is oxidized again by NAD+ to active lipoic acid and NADH. This step is catalyzed by dihydrolipoyl dehydrogenase (E3) encoded by *lpd*. Several coenzymes are also involved in the PDH reaction, including NAD+, coenzyme A, thiamine pyrophosphate, lipoic acid and FAD.

The complex is highly regulated by acetyl-CoA and NADH. High levels of either product allosterically inhibit the activity of PDH. PDH is also sensitive to the energy status of the cell. PDH is activated by AMP and inhibited by GTP.

#### 3.1.2. Pyruvate Formate-Lyase (PFL)

PFL encoded by *pflB* catalyzes the nonoxidative conversion from pyruvate and coenzyme A (CoA) to acetyl-CoA and formate under anaerobic conditions [24]:Pyruvate + CoA → Acetyl-CoA + Formate

This reaction plays an important role in anaerobic carbon metabolism in *E. coli* and other prokaryotes. PFL displays half-of-sites activity so that only one of the subunits is active at a time [24]. PFL is present in its inactive form under aerobic conditions, but in anaerobic conditions it is activated by an enzyme, which introduces a free radical into the inactive form of PFL. Studies showed that cells grown anaerobically expressed PFL protein 10 times more, when compared to cells grown aerobically [25]. The expression level of PFL protein can be further enhanced, by 1.5 to 2 times more, by adding pyruvate to the medium [25].

#### 3.1.3. Pyruvate Oxidase (POXB)

POXB is a flavoprotein that catalyzes the conversion of pyruvate to acetate and CO_2_, as shown below:
Pyruvate + H_2_O + FAD → Acetate + CO_2_ + FADH_2_

POXB was reported to perform a significant role during aerobic growth in glucose minimal medium [26]. Generally, acetyl units for biosynthetic purposes are provided by PDH aerobically and PFL anaerobically. POXB is considered to be essential during transition from stationary to exponential phases (through Acetyl-CoA synthetase), when PDH and PFL do not function fully [26].

This enzyme is a homo-tetramer and is encoded by the *poxB* gene. Each subunit contains a tightly, but non-covalently, bound FAD and loosely bound TPP through Mg^2+^ [26]. Phospholipids were reported to activate POXB strongly and to decrease Km for pyruvate by about 10 folds [26].

#### 3.1.4. Lactate Dehydrogenase (LDH)

There are three lactate dehydrogenases in *E. coli* which convert pyruvate to lactate [27]. Two of them (L-lactate dehydrogenase EC:1.1.1.27 and D-lactate dehydrogenase EC:1.1.1.28) are membrane bound (one specific for L-lactate and the other for D-lactate), couple to the respiration chain and are essential for aerobic growth. These two LDHs are also called lactate oxidases [27]. The fermentative LDH (encoded by *ldhA*) is a soluble enzyme which converts pyruvate to D-lactic acid [28], coupled to the consumption of NADH:
Pyruvate + NADH → Lactate + NAD+

The fermentative LDH of *E. coli* is induced under anaerobic conditions at acidic pH and is allosterically activated by pyruvate [28].

#### 3.1.5. PEP Synthase (PPS)

Phosphoenolpyruvate synthase (encoded by *PPS*) is responsible for converting pyruvate to PEP [29]:
ATP + Pyruvate + H_2_O → PEP + AMP + Phosphate

This enzyme is required for growth on single 3-carbon substrates, like pyruvate and lactate [29]. This reaction is a ping-pong reaction in which PPS is phosphorylated by ATP and, then, the phosphoryl group is transferred to pyruvate [30].

### 3.2. Biochemical Pathways Indirectly Impacting Pyruvate Formation

Acetyl-CoA Synthetase (ACS) and Acetyl-CoA (acetyl-CoA) are important metabolites in a variety of important physiological process that link anabolism with catabolism during growth on glucose. Another pathway which converts acetate to acetyl-CoA is catalyzed by acetyl-CoA synthetase (encoded by *ACS*). This enzyme is important to maintain adequate levels of acetyl-CoA, especially when organisms grow on acetate, or become unable to generate acetyl-CoA directly from pyruvate [31,32]. ACS belongs to a family of AMP-forming enzymes catalyzing a two-step reaction. The crystal structures of ACS suggest that the reaction mechanism of acetyl-CoA synthetase is a ping-pong reaction [31].

#### 3.2.1. PEP Carboxylase (PPC or PEPC. EC:4.1.1.31)

Phosphoenolpyruvate carboxylase encoded by the *PPC* gene is a CO_2_-fixing enzyme which catalyzes the conversion from PEP to oxaloacetate in the presence of Mg^2+^ [33]:$$\mathrm{PPC}+{{\mathrm{HCO}}_{3}}^{-}\;\overset{{\mathrm{Mg}}^{2+}}{\to \;}\mathrm{Oxaloacetate}\;+\;\mathrm{Phosphate}$$

PPC exists widely in a number of organisms, including bacteria, plants, and algae, but it is not found in animals, fungi, and yeasts.

During cell growth, four-carbon compounds, such as oxaloacetate, are withdrawn from the TCA cycle for biomass biosynthesis. These four carbon compounds need to be fed back into the TCA cycle from other reactions. *E. coli* PPC is activated by acetyl-CoA, fructose-1,6-bisphosphate, GTP, fatty acids and their CoA derivatives, and the enzyme is inhibited by L-aspartate, L-malate, citrate, succinate, and fumarate.

#### 3.2.2. Energy and (F1F0) H^+^-ATP Synthase Complex

Many studies have been conducted to address the question regarding the controls over flux through glycolysis and how this flux can be increased. In yeast, results suggested that no glycolytic enzyme exerted significant control over the flux pathway [34]. In *E. coli*, overexpressing enzymes that are helpful in glucose uptake and its phosphorylation did not impact the flux pathway [35]. Recent results demonstrated that controls of flux reside outside and not inside. Specifically, an increase in glycolytic pathway flow was observed in cells during energy depletion [36]. In *E. coli*, pyruvate kinase-II and phosphofructokinase-I were activated by AMP and ADP, respectively [37,38].

The proton-translocating (F1F0) H^+^-ATP synthase is located in the membrane in *E. coli* and performs oxidative phosphorylation [39]. This enzyme performs ATP synthesis by using electrochemical potential. The enzyme can also work in the opposite direction at the expense of ATP, when necessary, to generate a proton gradient, such as for locomotion, nutrient uptake, and other functions [40].

The soluble F1 subunit of (F1F0) H^+^-ATP synthase can hydrolyze ATP in vitro independently of the F0 subunit. The *E. coli pAGD* gene encoding α, β and γ chains of the F1 subunit of (F1F0) H^+^-ATP synthase was reported to exert the strongest ATPase activity [36]. The concentration of ATP in *E. coli* strain BOE270 was 25% lower than at the highest ATPase (EC:3.6.1.8) expression, and the concentration of ADP increased by 65%. The flux of the glycolytic pathway increased gradually with an increase of the ATPase expression and reached 170% of the wild-type flux at the highest expression level.

*E. coli TC36* (*adhE*, *atpFH*, *ldhA*, *frdBC*, *pflsucA*) was previously constructed for acetate production. To reduce growth during oxidative metabolism, mutations in (F1F0) H^+^-ATP synthase were introduced, which disrupted oxidative phosphorylation. Deletion of subunits b and c of F0 (encoded by *atpFH*) resulted in the separation of the F1 sector from the membrane. Therefore, the ATP synthesis function of (F1F0) H^+^-ATP synthase was disrupted, although the hydrolytic activity of F1-ATPase in the cytoplasm was preserved. The mutations resulted in a small reduction in cell yield and growth rate, compared with the wild-type and parent strains. A two-fold increase in glycolytic flux was observed, compared with *W3110* (wild type). This increasing flux was achieved due to increase in the ADP available for glycolysis and a decrease of allosteric inhibition of oxidative phosphorylation by ATP.

### 3.3. By-Products in the Production of Pyruvic Acid

Pyruvic acid is the key hub of the metabolic process in an organism, participating in many subsequent metabolic reactions and generating acetic acid, ethanol, lactic acid and other by-products. The production of these by-products not only consumes a large amount of pyruvic acid, but is also not conducive to the recovery and purification of the later products. Therefore, it is necessary to use metabolic engineering to reduce or prevent the generation of by-products.

Wang et al. knocked out the genes *pdc1* and *pdc5* encoding pyruvate decarboxylase in *S. cerevisiae Y2*, and optimized the fermentation medium. After 96 h in a shake flask of *S. cerevisiae Y2-15*, the yield of pyruvic acid increased 16.86 times, and the accumulated pyruvic acid increased to 24.65 g/L [41]. In another similar study, by knocking out the related gene encoding *PDC* in *T. glabra*, the accumulation of pyruvate in shake flask fermentation reached 20 g/L. It was 2.5 times higher than the original strain, while the production of the by-product, ethanol, was only 4.6 g/L. After 52 h fermentation in a fermentation tank, the yield of pyruvic acid was 82.2 g/L [42]. By deleting *ace E*, *pqo* and *ldhA* genes, acetyl hydroxy acid synthetase with reduced activity was introduced (*ΔC-TIlv N*). Constructed engineering of bacteria *C glutamicum ΔaceE Δpqo ΔldhA ΔC-tilv N* produced 190 mmol pyruvic acid under shaking flask conditions with a yield of 1.36 mol pyruvic acid/mol glucose, but still secreted a large amount of L-alanine. The mutant *alaT* and *avtA* genes, encoding alanine aminotransferase, were further knocked out. Under the condition of fed-batch high-density fermentation in a fermentation tank, the by-product L-valine reduced by 90%, and the yield of pyruvic acid was 0.97 mol pyruvic acid/mol glucose [43].

## 4. Production of Pyruvate by Recombinant Microbes

Although several microorganisms produce large quantities of pyruvate by using glucose and other carbon sources as substrates [44], *E. coli* and yeast are still preferred to produce pyruvate due to the simple nutritional requirements and rapid growth.

### 4.1. Production of Pyruvate by Recombinant E. coli

Use of genetic engineering strategies improved pyruvate accumulation in *E. coli* [45]. Although wild-type *E. coli* is not a commercially relevant producer of pyruvate, many strategies have been applied to modify wild-type *E. coli* metabolically to enhance its ability to accumulate the compounds. The strategies aim to enhance the flux flowing into pyruvate (e.g., glycolysis and LDH), and/or to repress or knock-out pathways which consume this compound (e.g., PDH and POXB). To increase pyruvate production, a number of *E. coli* strains have been studied (Table 1). An *E. coli* strain named *W1485lip2* (*lipA2*) was earlier used for pyruvate production under aerobic conditions, using glucose as the substrate in a medium lacking lipoic acid [46]. Decreased *pdh* activity in lipoic acid deficient conditions impaired conversion from pyruvate to acetyl coenzyme A. Under optimal culture conditions, by using 50 g/L glucose as the substrate, a total of 25.5 g/L pyruvate was obtained in 32–40 h at pH 6.0.

A *lipA2 bgl+ atpA401* mutant (*TBLA-1*) was constructed to produce pyruvate aerobically from glucose [47]. This F1-ATPase-defective mutant was constructed by transferring a defective gene, that was the alpha subunit of F1-ATPase, *atpA401*, into strain *W1485lip2*. *TBLA-1* produced more than 30 g/L of pyruvic acid from 50 g/L glucose in 24 h [47]. A decrease in the energy level of the *TBLA-1* strain enhanced the glycolytic pathway flow and, thereby, pyruvate production. *TBLA-1* consumed more oxygen and possessed higher b-type cytochromes content than its parent strain (*W1485lip2*). The activities of the PTS for glucose uptake were higher in *TBLA-1* than in W1485lip2. During exponential growth, a higher level of pyruvate kinase I and phosphoglycerate kinase was reported in *TBLA-1* compared to *W1485lip2*.

An *aceF* mutant (*CGSC6162*) and an *aceFppc* mutant (*CGSC7916*) were studied in complex media by using glucose and acetate as carbon sources. More than 30 g/L pyruvate was obtained with both *CGSC6162* and *CGSC7916* [22]. High mass yields were achieved in *CGSC6162* (0.72) and *CGSC7916* (0.78). In *CGSC6162* and *CGSC7916,* the volumetric productivities were 1.5 g/(L·h) and 1.2 g/(L·h), respectively. The higher pyruvate production rate occurred early in fermentation. The studies of the effect of pH and temperature on production of pyruvate showed that at 32 °C and pH 7.0, the pyruvate production was higher.

An *E. coli* strain *YYC202 ldhA::Kan* (*Hfrzbi::Tn10 poxB1 ΔaceEF) rpsL pps-4 pfl-1 ldhA*) was used for the production of pyruvate in fed-batch fermentation [48]. This auxotroph strain needs acetate for growth in a glucose minimal medium, due to the complete inhibition of the conversion from pyruvate to acetyl-CoA. A final pyruvate concentration higher than 62 g/L, a productivity of 1.75 g/(L·h) and a pyruvate/glucose molar yield of 1.11 mol/mol (0.56 g/g) were obtained. At the optimal process conditions, the acetate feeding rate was regulated by an open-loop acetate control system, based on a correlation between volume-specific CO_2_ transfer rate and acetate consumption rate. In a subsequent study, using intergraded electrodialysis, a volumetric productivity of 4.5 g/(L·h) was achieved [32]. However, due to apparent strain instabilities, the fermentation time was limited to about 40 h. Using a repetitive fed-batch process, a yield of 0.89 g/g and a productivity of 6.04 g/(L·h) were obtained. By a fully integrated approach, a (calculated) final pyruvate concentration of 79 g/L was achieved.

A strain, *TC44*, was used to produce pyruvate in a mineral salt medium with glucose as the carbon source [49]. The *pfl*, *ldh*, and *poxB* encoding alcohol dehydrogenase, acetate kinase, and fumarate reductase, respectively, were knocked out to prevent pyruvate from converting into other fermentation products. In this strain, the glycolytic flux was 50% more than that of the *W3110* strain (the parental strain). This increase in flux was due to the increase in ADP availability for glycolysis and the decrease in ATP accumulation by introducing the ATPase mutation (*ΔatpFH*). The oxidative phosphorylation was disrupted in this strain, while the hydrolytic activity of F1-ATPase was preserved. A principal difference between this strain and other pyruvate production strains was the presence of PDH. The TCA cycle was interrupted by repressing *sucAB* encoding 2-ketoglutarate dehydrogenase (EC:1.2.4.2) instead of *PDH* knock out to prevent pyruvate from being completely oxidized to CO_2_. This approach achieved 67.4 g/L with a yield of 0.75 g/g and a productivity of 1.2 g/(L·h) in a glucose defined medium. microorganisms-10-02454-t001_Table 1Table 1Recombinant *E. coli* strains used for the production of pyruvate.StrainMutationsAuxotrophYield(g/g)Pyruvate(g/L)Productivity(g/L·h)MediaReference*W1485lip2*
Lipoic acid0.5125.50.64Complex[46]*TBLA-1**atpA401*Lipoic acid0.6030.01.25Complex[47]*CGSC6162**aceF*
0.72>30.01.50Complex[22]*CGSC7916**aceF*, *ppc*
0.78>30.01.20Complex[22]*E. coliTC44*

0.7565.96

[49]*E. coli CGSC6162 Deltappc*

0.7835

[22]*E. coli ALS929*

0.6890

[50]*E. coli YYC202 ldhA::Kan*

0.87110.0

[32]*YYC202**ldhA::Kan**aceEF*,*ldhA*, *pps*, *pfl*, *poxB*
0.5662.01.75
[48]*TC44**adhE*, *atpFH*, *ackA*, *ldhA*, *frdBC*, *pfl*, *poxB*, *sucA*
0.7567.41.20
[49]


### 4.2. Production of Pyruvate by Recombinant Yeast

Yeast is the preferred microbe for pyruvate production, using different types of substrates as carbon sources. For example, *Candida lipolytic AJ 14353* accumulated 43.6 g/L pyruvate at a yield of 0.44 g/g in 72 h and *S. cerevisiae* accumulated 36.9 g/L pyruvate at a yield of 0.37 g/g in 48 h under conditions of thiamine limitation. Van Mari et al. bred a strain of *S. cerevisiae TAM* which was independent of C2 compound and glucose tolerant and lacked pyruvate decarboxylase. When the initial glucose concentration of *TAM* was 100 g/L, the pH was controlled at 5.0. After 60 h of inoculation, the pyruvate content reached 100 g/L, and the final pyruvate concentration reached 135 g/L after 100 h [51].

These strains are thiamine auxotrophic. Thiamine acts as a cofactor for pyruvate decarboxylase and PDH complex. A multi-vitamin auxotroph of *Torulopsis glabrata* accumulated about 68 g/L in 63 h at a yield of 0.494 g/g. Thiamine, nicotinic acid, biotin and pyridoxine were required in the medium to grow this strain. These vitamins are co-factors of PDH, pyruvate decarboxylase, pyruvate carboxylase, and transaminase, respectively. Accumulation of pyruvate by these strains was caused by the reduced activity of these enzymes under conditions of deficiency of these vitamins. We recently reported that glycerol was a promising substrate for *Y. lipolytica* to produce pyruvate [52]. Furthermore, we observed an enhanced production of pyruvate from an osmotic stress-resistant mutant of *Y. lipolytica* [52].

Kamzolova, S et al. constructed a thiamine deficient yeast by disturbing thiamine-dependent pyruvate dehydrogenase. Finally, during cultivation in a fermenter, the strain *Blastobotrys adeninivorans* VKM Y-2677 produced 43.2 g/L PA from glucose with a product yield (YPA) of 0.77 g PA/g glucose. The proportion of PA to byproducts was 18:1 for KGA and 8:1 for citric acid [53]. There are two glycerol decomposition pathways in *S. cerevisiae.*, namely the dihydroxyacetone pathway (involving *GCY1, DAK1, DAK2*) [54] and glycerol-3-phosphate pathway (involving *GUT1* and *GUT2*) [55,56]. Over-expression of *GCY1* and *DAK1* in *S. cerevisiae* enhanced glycerol decomposition in [57]. Similarly, overexpression of *GUT1* and *GUT2* was reported to enable effective glycerol assimilation toward the synthesis of desired products in *Y. lipolytica.* Earlier, we observed that glycerol was the better carbon source for producing pyruvate in *Y. lipolytica*, compared to glucose and other carbon sucrose [52]. Glucose and glycerol are the main carbon sources for producing pyruvate from engineered microbes (Table 2).

Miyata found that some nutrient deficient yeasts of vitamin battalion are beneficial to the accumulation of pyruvate, such as *T. glabrata IF0005* (Na^+^, BI^−^, B^−^, Bio^−^). The production of pyruvate reached 480 mml/L after 135 h of fermentation at the initial glucose concentration of 100 g/L, 60% higher than that of the original strain [58].

KGA was excreted from yeast cells as a byproduct of PA over-synthesis. Furthermore, the increased level of KGA in the yeast cells inhibited NAD-dependent isocitrate dehydrogenase in the TCA cycle and enhanced the production and excretion of citric acid, another byproduct of PA over-synthesis. The proportion of PA to by-products approximate was 18:1 for KGA and 8:1 for citric acid [53].

The overall metabolic direction of cells can be predicted by analyzing metabolism, but the metabolism of microorganisms in vivo is complex, and the specific metabolism needs to be precisely located. Stable isotope labeling is a feasible method. Maaheimo H et al. analyzed the carbon source metabolism of *S. cerevisiae* under low glucose concentration (aerobic and anaerobic conditions) through BDF ^13^C labeling NMR and pointed out that the pentose phosphate pathway in cells only provided biosynthesis and did not participate in the energy supply system. In the aerobic system, 75% of mitochondrial pyruvate came from the pyruvate recycling pathway. The TCA cycle was mainly used for biosynthesis [59]. Fiaux J et al. compared the metabolic flow of glucose between *S. cerevisiae* and *Pichia stipitis* through BDF ^13^C labeling NMR, and quantified the proportion of the metabolic flow direction of *S. cerevisiae* overflow [60]. Therefore, in order to reduce the generation of metabolic by-products, we can analyze the metabolic pathway through isotope labeling and other methods.

Recently, we used CRISPR/Cas9 technology to knock out the *OACLP* gene, encoding the oxaloacetate transporter, preventing pyruvate from entering the mitochondria and being consumed by the TCA cycle in *Y. lipolytica*. Since high concentrations of pyruvate affect pyruvate synthesis, ATP consumption was constructed by heterologous expression of the *Pyc2* encoding pyruvate decarboxylase and *PCK1* encoding phosphoenolpyruvate carboxylase from *S. cerevisiae* in *Y. lipolytica*. Finally, the pyruvic acid production in the 5-L fermenter reached 96.9 g/L, which was 41.6% higher than the starting strain constant.
microorganisms-10-02454-t002_Table 2Table 2Yeast and other microorganisms for the production of pyruvate.StrainSubstratePyruvate (g/L)Yield (g/g)Details and References*Bacillus megaterium*glucose27.80.38[61]*Blastobotrys adeninivorans VKM Y-2677*glucose43.20.77[53]*S. cerevisiae*glucose1350.54[51]*Trichosporoncutaneum PD70*glucose34.60.429[62]*Torulopsis glabrata*glucose94.30.635[63]*Y. lipolytica**a 374/4*glycerol61.30.71*Y. lipolytica*[64]*Y. lipolytica**a WSH-Z06*glycerol39.30.71[65]*Y. lipolytica**a*glycerol48.10.48[66]*Y. lipolytica**a*glycerol97.20.795[52]


## 5. Summary

The fundamental aim of this review was to understand aspects of the regulation of microbial central metabolism and use metabolic engineering tools and fermentation process development to maximize the production of pyruvate. Wild-type microorganisms normally are not able to accumulate a single compound at a high rate and a high concentration if metabolism is not tightly regulated. The development of recombinant DNA technology provides a new way for the directed improvement of product yield and productivity. For example, by deleting pathways leading to by-products, the flow of the substrate can be redirected toward products. Fermentation process development is also essential for the improvement of product formation, and altering pathways can affect the process conditions, such as pH, temperature, oxygenation, and ion strength. By further understanding the inhibition mechanism of high osmotic stress and ion strength on cell growth and pyruvate production, additional genetic modifications might be able to increase tolerance of a strain in high ion strength.

As an intermediate in the synthesis of many drugs and pesticides, pyruvic acid has a very broad market prospect. However, there is a big gap in the domestic and foreign markets for pyruvic acid, especially for high-quality pyruvic acid. The traditional tartaric acid synthesis has its natural disadvantages. In recent years, there has been rapid development in the field of bioengineering, especially the emergence of gene editing technology, represented by CRISPR/Cas9, which has given more options to modifying microorganisms at the genetic level. Some yeasts, such as *Y. lipolytica*, have high homologous recombination efficiency. They are relatively easy to perform gene editing operations on so as to facilitate heterologous expression of related genes. In this way, some microorganisms can better combine the natural metabolic pathway of pyruvate. In becoming a natural strain for pyruvic acid production, this is more conducive to improving the production efficiency of pyruvic acid. In addition, with the cross development of genetic engineering, molecular biology, systems biology and other disciplines, rational research on microbial metabolic pathways and gene expression regulation networks has been promoted. Metabolic engineering is also gradually turning towards the development of dynamic regulation, such as cell self-induced dynamic regulation, through metabolite response elements or quorum sensing elements, or artificial control of physical or chemical signals to stimulate promoters and other elements to regulate the expression of downstream genes, providing more reliable theoretical support and technical means for dynamic regulation of microbial fermentation to produce pyruvate.

It is undeniable that, although the production of pyruvic acid by microorganisms is a major trend, there are still many deficiencies in the biological production of pyruvic acid. First, the number of food grade microorganisms that can be used to produce pyruvic acid is far from enough. Secondly, although biotechnology has made some progress in recent years, when using this technology to transform microorganisms, these technologies still cannot achieve the expected effects, which greatly limits the transformation of specific strains. In addition, microbial fermentation is affected by many factors, which affect the final pyruvic acid production. Therefore, it is hoped that more safe microorganisms can be found in the future, and more technologies for microbial transformation can be developed, so that the microbial production of pyruvic acid can be commercialized on a large scale at an early date.

## Figures and Tables

**Figure 1 microorganisms-10-02454-f001:**
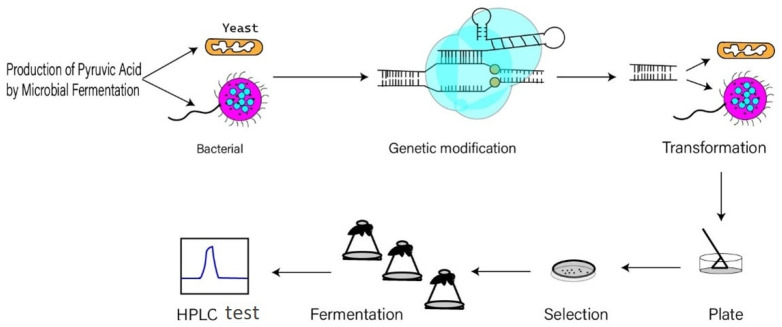
Pyruvic acid production process by microbial fermentation.

**Figure 2 microorganisms-10-02454-f002:**
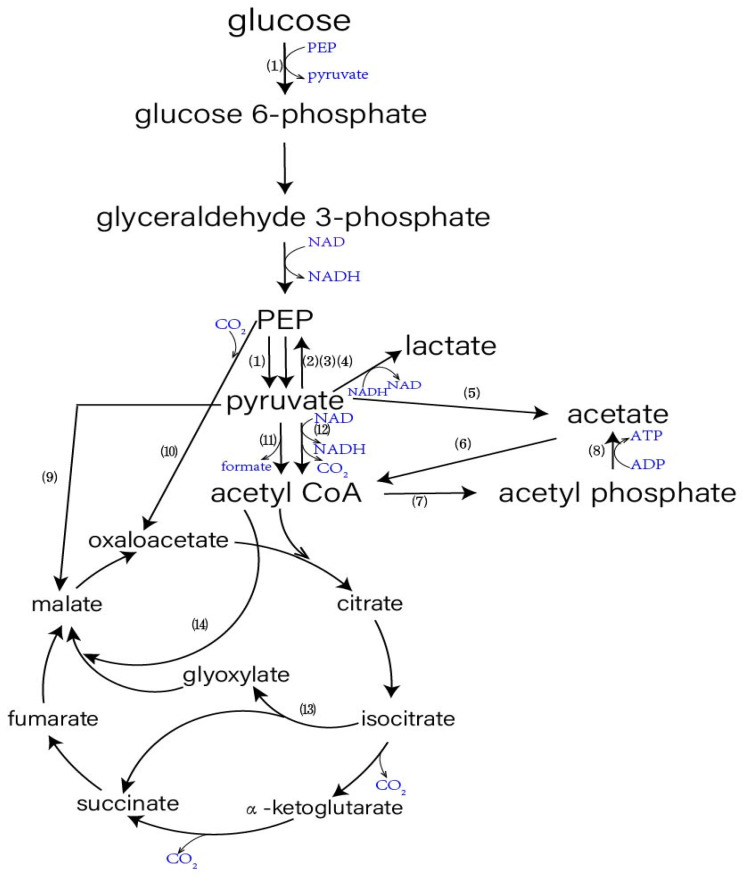
Metabolic pathway of *E. coli* related to pyruvate production. (1) phosphotransferase system (PTS). (2) pyruvate oxidase (PYK). (3) PEP synthase (PPS). (4) lactate dehydrogenase (LDH). (5) pyruvate oxidase (PoxB). (6) acetyl CoA synthetase (ACS). (7) phosphotransacetylase. (8) acetate kinase. (9) malic enzyme (ME). (10) PEP carboxylase (PPC). (11) pyruvate formate-lyase (PFL). (12) pyruvate dehydrogenase complex (PDH). (13) isocitrate lyase. (14) malate synthase.

## Data Availability

Not applicable.

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
