# Peer review of "The Production of Pyruvate in Biological Technology: A Critical Review"

_microorganisms, 2022, doi:10.3390/microorganisms10122454_

Round 1

Reviewer 1 Report (New Reviewer)

I cannot recommend the work for publication in this form because it is too carelessly written.

Details:

The content is repeated in the introduction.

No page numbering or line numbering.

What year is the information on the cost of producing pyruvate from?

In the fermentation chapter-fragment, you are not naming any specific yeast and E. coli strains while referring to some unspecified ones.

Why do the authors capitalize or lowercase the word yeast?

Like the names of enzymes, numbers for enzymes are also missing.

Figure 1 is illegible on the printout, enzyme numbers are not visible, and the diagram contains errors. Lots of punctuation errors throughout the manuscript.

The subsection title should start with a capital letter.

The species name is Yarrowia lipolytica and not Yarrowia lipolytic.

In tables: g/L or g/l?

Antonius J. A. van Mari et al. - what is this position?

The list of literature is unacceptable - each item is spelled differently and the names of journals as well, species names of microorganisms should be written in italics.

Publication number 67 is misspelled, it should be written:

Cybulski, K.; Tomaszewska-Hetman, L.; Rakicka, M.; Łaba, W.; Rymowicz, W.; Rywińska, A. The bioconversion of waste products from rapeseed processing into keto acids by Yarrowia lipolytica. Industrial Crops and Products 2018, 119, 102-110, doi:10.1016/j.indcrop.2018.04.014

Author Response

Dear reviewer:

   Thank you very much for your valuable suggestions on the revision of my article. I have revised it according to your requirements.

Point 1: The content is repeated in the introduction.

Response 1: Thank you very much for your comments. I have deleted the repeated parts in the introduction and revised the introduction again.

Point 2: No page numbering or line numbering.

Response 2: I've added the page number of the article.

Point3: What year is the information on the cost of producing pyruvate from?

Response 3: I have calculated the raw material consumption required for pyruvic acid production according to the latest market price.

Point 4: Why do the authors capitalize or lowercase the word yeast?

Response 4: Thank you very much for your reminding. I have solved this problem and modified other syntax problems in the article.

Point 5: Like the names of enzymes, numbers for enzymes are also missing.

Response 5: Thank you very much for your reminding. I have modified the names and numbers of the enzymes involved.

Point 6: Figure 1 is illegible on the printout, enzyme numbers are not visible, and the diagram contains errors. Lots of punctuation errors throughout the manuscript.

Response 6: Thank you very much for your reminding. I have modified Figure 1.

Point 7: The subsection title should start with a capital letter.

Response 7: I have checked each sub title and made changes

Point 8: The species name is Yarrowia lipolytica and not Yarrowia lipolytic.

Response 8: Thank you very much for reminding me that I have modified the names of all strains.

Point 9: In tables: g/L or g/l?

Response 9: Thank you very much. I have corrected this problem.

Point 10: Antonius J. A. van Mari et al. - what is this position?

Response 10: Thank you very much. I have corrected this problem.

Point 11: The list of literature is unacceptable - each item is spelled differently and the names of journals as well, species names of microorganisms should be written in italics.

Response 11: Thank you very much. I have solved the problems in the references.

Point 12: Publication number 67 is misspelled, it should be written:

Cybulski, K.; Tomaszewska-Hetman, L.; Rakicka, M.; Łaba, W.; Rymowicz, W.; Rywińska, A. The bioconversion of waste products from rapeseed processing into keto acids by Yarrowia lipolytica. Industrial Crops and Products 2018, 119, 102-110, doi:10.1016/j.indcrop.2018.04.014

Response 12: Thank you very much for finding this problem. I have revised this reference according to your format.

Reviewer 2 Report (New Reviewer)

The authors have focused on bio-based production of pyruvate. The review is well written and organized. Here are a few comments for authors to consider>

a) The title seems inappropriate since the authors mainly focused on two hosts E. coli and Yeast. 

b) Yeast is a very general term. The majority of review focuses on pyruvate production in S. cerevisiae, whereas a few others are just superficially touched. I will recommend adding more detail for other yeast.

c) References are missing at multiple places. For instance Page 2"current research shows that the microorganisms that ferment......", No refrence for chemical method section.

d) Although the authors discussed that the majority of pyruvic acid is produced using chemical methods and mentioned limitations also. But they should also mention the limitations for biological methods why it is not popular for pyruvic acid compare to chemical methods.

e) Table 1 has E. coli strains. Table 2 has mix E. coli and Yeast strains. Why not to make one table for bacterial strains for pyruvic acid bio production and the second table for Yeast strains.

f) I will recommend to add more figures for easy understanding of what is mentioned in text. The authors can make a figure highlighting bacterial and yeast strategies for pyruvic acid production with respective bottlenecks / troubleshooting.

g) There are a few spelling / grammatical errors. For instance Page 2 "chemical sectors to synthesis other important goods", Page 4 "Therefore it's are called the general PTS protein.....". Please check throughout the manuscript.

Author Response

Dear reviewer

      Thank you very much for your valuable comments on my article and let me further improve it. I have revised my article according to your requirements.

Point 1: The title seems inappropriate since the authors mainly focused on two hosts E. coli and Yeast.

Response 1: Thank you very much for your comments. The main purpose of this documentary is to show the readers the advantages and methods of microbial fermentation to produce pyruvic acid and the shortcomings of current biological transformation technology through bacteria and yeasts such as Escherichia coli. We are sorry for your misunderstanding.

Point 2: Yeast is a very general term. The majority of review focuses on pyruvate production in S. cerevisiae, whereas a few others are just superficially touched. I will recommend adding more detail for other yeast.

Response 2: Thank you very much for your suggestion. I reviewed the literature and added the following parts to the article:

            Kamzolova, S et al constructed a thiamine deficient yeast by disturbing thiamine-dependent pyruvate dehydrogenase. Finally, during cultivation in a fermentor, the strain Blastobotrys adeninivorans VKM Y-2677 produced 43.2 g/l PA from glucose with a product yield (YPA) of 0.77g         ② PA/g glucose. The proportion of PA to byproducts was 18:1 for KGA and 8:1 for citric acid.

In the article, I also mentioned our recent method of synthesizing pyruvate in Yarrowia lipolytica

Point3: References are missing at multiple places. For instance Page 2"current research shows that the microorganisms that ferment......", No refrence for chemical method section.

Response 3: I am very sorry for this problem. I have supplemented the part of "chemical synthesis of pyruvic acid" and added references. The format of references has been modified.

Point 4: Although the authors discussed that the majority of pyruvic acid is produced using chemical methods and mentioned limitations also. But they should also mention the limitations for biological methods why it is not popular for pyruvic acid compare to chemical methods.

Response 4: Thank you very much for your suggestion. At the end of the article, I have supplemented the deficiency of pyruvic acid synthesis by biological method.

Point 5: Table 1 has E. coli strains. Table 2 has mix E. coli and Yeast strains. Why not to make one table for bacterial strains for pyruvic acid bio production and the second table for Yeast strains.

Response 5: Thank you very much for your suggestion. I have rearranged the two tables.

Point 6:  I will recommend to add more figures for easy understanding of what is mentioned in text. The authors can make a figure highlighting bacterial and yeast strategies for pyruvic acid production with respective bottlenecks / troubleshooting.

Response 6: I added the flow chart of pyruvic acid production by microbial fermentation to the article. I have added the current shortcomings of this method to the end of the article.

Point 7: There are a few spelling / grammatical errors. For instance Page 2 "chemical sectors to synthesis other important goods", Page 4 "Therefore it's are called the general PTS protein.....". Please check throughout the manuscript.

Response 7: Thank you very much for reminding me. I have rechecked the whole article and corrected the syntax errors in the article.

Round 2

Reviewer 1 Report (New Reviewer)

I accept the manuscript in present form

This manuscript is a resubmission of an earlier submission. The following is a list of the peer review reports and author responses from that submission.

Round 1

Reviewer 1 Report

Wei Yuan and co-workers composed a review addressing the production of pyruvate in different microbial hosts. Although the authors expanded the amount of relevant information beyond a previously published review on “Biotechnological routes to pyruvate production” by Ping Xu et al. (2008; DOI: 10.1263/jbb.105.169), the submitted manuscript could be improved on various levels.

The abstract already contains repeated – in other words: redundant or only slightly modified – information, for example, the higher costs of chemical pyruvate synthesis and the advantages of fermentative processes employing (engineered) microorganisms. Repeated/redundant information is flaw present throughout all sections of review and should be critically improved. Further, the abstract specifically highlights yeast, suggesting that the review will focus on yeasts, which is actually not the case. Re-phrasing is strictly recommended.

The two sections after the introduction addresses biochemical pathways directly/indirectly influencing pyruvate yields in vivo. To improve this section beyond just a list of biochemical (textbook) reactions, the authors are asked to implement and describe how scientists have used metabolic engineering to reroute fluxes targeting these reactions and reference appropriate examples. This section could also be improved by balancing the amount of information. Some subsections brim over with details (e.g., size of protein subunits), whereas other sections lack information, for example, the description of uncommon terms (e.g., Bi-Uni-Uni-Bi ping-pong reaction).

In the section “General methods of pyruvic acid production” the reference style sometimes changes from numbers in rectangular brackets [NUMBER] to author and year in round brackets (First author et al., 2000). The latter references are not found in the reference list at the end of the review, hence, are missing. Regarding citations, the authors cite their own publication, which is appropriate in the context of the review, but the cite the same paper multiple times: references [47], [48], and [62] are identical. Also [24] and [25] are identical. The authors have to correct this. The section on the chemical methods does not include yields of pyruvate, which makes it difficult to put it into perspective to the biotechnological methods. Essential experimental details should be added wherever suitable for clarity and understanding.

Further improvements should aim at but are not limited to:

The font changes in different sections, for example, “Biochemical pathways indirectly impacting pyruvate formation”, the first paragraph in “Biotechnological methods”, and “Production of pyruvate by recombinant E. coli”. Is this due compiling the manuscript from multiple documents?

The authors are asked to check the whole document for typos, spacing, paragraphs, general formatting (e.g., gene names should be italic and not capitalized; microbial strains like E. coli should be italic and capitalized), consistent use of units, introduction and subsequent use of abbreviations etc.

Correct referencing has been addressed earlier and has to be revised. The reference in the first entry of Table 2, for example, is certainly not [53].

The submitted review by W. Yuan et al. needs in depth revision and careful contextualization of the field. Further, it is hard to grasp future trends from the final summary, which recaps mostly established concepts. All in all, publication should only be considered after extensive revision.

Author Response

  First of all, I am very honored that you can review my article. After seeing your suggestions for revision, I have re-corrected the upcoming article, as follows:

Point 1:The abstract already contains repeated – in other words: redundant or only slightly modified – information, for example, the higher costs of chemical pyruvate synthesis and the advantages of fermentative processes employing (engineered) microorganisms. Repeated/redundant information is flaw present throughout all sections of review and should be critically improved. Further, the abstract specifically highlights yeast, suggesting that the review will focus on yeasts, which is actually not the case. Re-phrasing is strictly recommended.

Response1:

Many thanks for your constructive suggestions. After consideration, I agree with you. According to your opinion, I have cut out the relatively repeated parts without affecting the reader's reading. The pyruvate-producing microorganisms described in this review are not only yeast but also microorganisms such as Escherichia coli.

Point 2: The two sections after the introduction addresses biochemical pathways directly/indirectly influencing pyruvate yields in vivo. To improve this section beyond just a list of biochemical (textbook) reactions, the authors are asked to implement and describe how scientists have used metabolic engineering to reroute fluxes targeting these reactions and reference appropriate examples. This section could also be improved by balancing the amount of information. Some subsections brim over with details (e.g., size of protein subunits), whereas other sections lack information, for example, the description of uncommon terms (e.g., Bi-Uni-Uni-Bi ping-pong reaction).

Response2:

Thank you very much for your reminder, I have corrected the unreasonable technical terms in the article. I also add some examples of how scientists balance metabolic fluxes through metabolic engineering. For example, heat treatment of M. roberii increases pyruvate kinase activity and thereby increases pyruvate production. Likewise, downregulation of genes such as MAA `_02871 and MAA_08787 enhanced pyruvate production. Mutation of phosphoenolpyruvate carboxylase (ppc), pyruvate dehydrogenase (aceF), and alcohol dehydrogenase (adhE) in E. coli enhanced pyruvate production. In addition, thiamine or lipoic acid auxotrophs have been found to accumulate pyruvate. Auxotrophs grown in lipoic acid- or thiamine-limited media reduced the specific activity of the pyruvate dehydrogenase complex (PDHc), thereby enhancing pyruvate accumulation.

Point 3: In the section “General methods of pyruvic acid production” the reference style sometimes changes from numbers in rectangular brackets [NUMBER] to author and year in round brackets (First author et al., 2000). The latter references are not found in the reference list at the end of the review, hence, are missing. Regarding citations, the authors cite their own publication, which is appropriate in the context of the review, but the cite the same paper multiple times: references [47], [48], and [62] are identical. Also [24] and [25] are identical. The authors have to correct this. The section on the chemical methods does not include yields of pyruvate, which makes it difficult to put it into perspective to the biotechnological methods. Essential experimental details should be added wherever suitable for clarity and understanding.

Response3:

Thank you very much for your reminder, I have corrected the problem of using different serial numbers for the same document. For citing my own literature, this is entirely out of the needs of this literature and not intentionally done. I hope the reviewers can understand it. I have also made changes to the changes in the reference format you mentioned.

Point 4:The font changes in different sections, for example, “Biochemical pathways indirectly impacting pyruvate formation”, the first paragraph in “Biotechnological methods”, and “Production of pyruvate by recombinant E. coli”.

The authors are asked to check the whole document for typos, spacing, paragraphs, general formatting (e.g., gene names should be italic and not capitalized; microbial strains like E. coli should be italic and capitalized), consistent use of units, introduction and subsequent use of abbreviations etc.

Response4:

Thank you very much, all the formatting issues such as paragraphs, fonts, gene abbreviations italics, and microorganism nouns italics have been revised. I'm very sorry for this relatively low-level question, thank you very much for pointing it out

Point 5: Correct referencing has been addressed earlier and has to be revised. The reference in the first entry of Table 2, for example, is certainly not [53].

Response5:

I have reworked the references to correspond to the original.

Point 6:The submitted review by W. Yuan et al. needs in depth revision and careful contextualization of the field. Further, it is hard to grasp future trends from the final summary, which recaps mostly established concepts.

Response6:

I'm very sorry for the absence of this part of a review that should have added my own views on the future at the end. I have combined the current production of pyruvate and my own views and added my own future prospects for the use of metabolic engineering and gene editing tools to transform microorganisms to produce pyruvate at the end of the article.

Reviewer 2 Report

On the manuscript entitled "The production of pyruvate in biological technology” for Microorganisms

The subject is of interest for the readers of Microorganisms. However, in my opinion the manuscript is not well organized.

Point 1: In the Introduction, it is necessary to explain what the aim of this review.

Point 2: After the Introduction, it is better to present the material in this order – the methods for pyruvic acid production, the producers, and conditions for the pyruvic acid synthesis with the indication of the key factors for bacterial and yeast strains, the biochemistry of the pyruvic acid production in bacteria and yeast.

Point 3: It is necessary to discuss the problem of the synthesis of by-products when using various carbon sources and microorganisms, in particular, α-ketoglutaric acid when using yeast producers.

Point 4: For enzymes, the EC numbers must be indicated.

Point 5: It is necessary to present a scheme for the production of pyruvic acid by yeast.

Author Response

  First of all, I am very honored that you can review my article. After seeing your suggestions for revision, I have re-corrected the upcoming article, as follows:

Point 1: In the Introduction, it is necessary to explain what the aim of this review.

Response1:

Thank you very much for the reminder, I have added the purpose of this review at the end of the introduction.

Point 2: After the Introduction, it is better to present the material in this order – the methods for pyruvic acid production, the producers, and conditions for the pyruvic acid synthesis with the indication of the key factors for bacterial and yeast strains, the biochemistry of the pyruvic acid production in bacteria and yeast.

Response2:

After your reminder, I have adjusted the order of each chapter of the article.

Point 3: It is necessary to discuss the problem of the synthesis of by-products when using various carbon sources and microorganisms, in particular, α-ketoglutaric acid when using yeast producers.

Response3:

Thank you. After seeing this modification suggestion, I reviewed the relevant literature to summarize the by-products in the production of pyruvate, and finally put this part at the end of the section "Production of pyruvate by recombinant Yeast".

Point 4: For enzymes, the EC numbers must be indicated.

Response4:

Thank you, I have added the EC numbers to the article.

Point 5: It is necessary to present a scheme for the production of pyruvic acid by yeast.

Response5:

Thanks a lot for your suggestion, I've added a strategy for pyruvate production by yeast in the section “Production of pyruvate by recombinant Yeast”.

Reviewer 3 Report

These authors globally reviewed the production of pyruvate using biological technology, and the main enzymes and pathways were summarized. Moreover, the metabolic engineering strategies to enhance pyruvate production in E. coli and Yeast were described. 

This review covers the development of the production of pyruvate using biotechnology. 

A few points should be fixed in this review. 

1, the title of the review should be further polished. 

2, when the full name appeared in the manuscript secondly, it should be abbreviated. Some microbial names, such as Escherichia coli, Saccharomyces cerevisiae, Yarrowia lipolytica, are not abbreviated.

3, Species names should be italic, many are not. 

4, Except summarizing current advances of pyruvate production, future perspective should be added. Especially, the synthetic biology strategy should be discussed. 

5, The word style is not constant in the manuscript, the authors need to fix. 

Author Response

Point 1 the title of the review should be further polished. 

Response:

For the title of the article, you may feel that the title of the article needs further revision due to various problems in the article. I later modified the questions that appeared in the article to suit this topic, hoping to meet your requirements.

Point 2 when the full name appeared in the manuscript secondly, it should be abbreviated. Some microbial names, such as Escherichia coli, Saccharomyces cerevisiae, Yarrowia lipolytica, are not abbreviated.

Point 3Species names should be italic, many are not. 

Point 5The word style is not constant in the manuscript, the authors need to fix. 

Response:

Thank you very much, all the formatting issues such as paragraphs, fonts, gene abbreviations italics, and microorganism nouns italics have been revised.

Point 4 Except summarizing current advances of pyruvate production, future perspective should be added. Especially, the synthetic biology strategy should be discussed. 

Response:

I'm very sorry for the absence of this part of a review that should have added my own views on the future at the end. I have combined the current production of pyruvate and my own views and added my own future prospects for the use of metabolic engineering and gene editing tools to transform microorganisms to produce pyruvate at the end of the article.

Round 2

Reviewer 1 Report

Original review round P1:

Wei Yuan and co-workers composed a review addressing the production of pyruvate in different microbial hosts. Although the authors expanded the amount of relevant information beyond a previously published review on “Biotechnological routes to pyruvate production” by Ping Xu et al. (2008; DOI: 10.1263/jbb.105.169), the submitted manuscript could be improved on various levels.

The abstract already contains repeated – in other words: redundant or only slightly modified – information, for example, the higher costs of chemical pyruvate synthesis and the advantages of fermentative processes employing (engineered) microorganisms. Repeated/redundant information is flaw present throughout all sections of review and should be critically improved. Further, the abstract specifically highlights yeast, suggesting that the review will focus on yeasts, which is actually not the case. Re-phrasing is strictly recommended.

Response P1 by the authors:

Many thanks for your constructive suggestions. After consideration, I agree with you. According to your opinion, I have cut out the relatively repeated parts without affecting the reader's reading. The pyruvate-producing microorganisms described in this review are not only yeast but also microorganisms such as Escherichia coli.

Review 2 to response P1:

The abstract has been slightly modified. Whereas the addition of E. coli is suitable, why not referring to a more general phrasing like “microorganisms such as different yeasts or E. coli” since the review also includes studies with Bacillus strains, for example? What has not been addressed is the inclusion of the previous review by Ping Xu et al. (2008; DOI: 10.1263/jbb.105.169). I think, this would be valid since a good amount of information in Table 1 and Table 2 has already been reviewed before. Further, the titles of the two tables “Recombinant E. coli strains used for pyruvate production” and “Pyruvate production from recombinant microbes” should be improved; Table 2 also features E. coli besides other organisms and both titles claim to highlight recombinant microorganisms.

Original review round P2:

The two sections after the introduction addresses biochemical pathways directly/indirectly influencing pyruvate yields in vivo. To improve this section beyond just a list of biochemical (textbook) reactions, the authors are asked to implement and describe how scientists have used metabolic engineering to reroute fluxes targeting these reactions and reference appropriate examples. This section could also be improved by balancing the amount of information. Some subsections brim over with details (e.g., size of protein subunits), whereas other sections lack information, for example, the description of uncommon terms (e.g., Bi-Uni-Uni-Bi ping-pong reaction).

Response P2 by the authors:

Thank you very much for your reminder, I have corrected the unreasonable technical terms in the article. I also add some examples of how scientists balance metabolic fluxes through metabolic engineering. For example, heat treatment of M. robertsii increases pyruvate kinase activity and thereby increases pyruvate production. Likewise, downregulation of genes such as MAA_02871 and MAA_08787 enhanced pyruvate production. Mutation of phosphoenolpyruvate carboxylase (ppc), pyruvate dehydrogenase (aceF), and alcohol dehydrogenase (adhE) in E. coli enhanced pyruvate production. In addition, thiamine or lipoic acid auxotrophs have been found to accumulate pyruvate. Auxotrophs grown in lipoic acid- or thiamine-limited media reduced the specific activity of the pyruvate dehydrogenase complex (PDHC), thereby, enhancing pyruvate accumulation.

Review 2 to response P2:

Thank you for partly editing the information content and the addition of this paragraph to the section “Biochemical pathways directly impacting pyruvate”. However, my main criticism was – and still is – that the chapter “Biochemical pathways involved in pyruvic acid” – I would change it to “Biochemical pathways involving pyruvate”. – including a list of reactions are not put into context critically enough with the next chapter “Production of pyruvate from recombinant microbes”. – I would change it to “Production of pyruvate by recombinant microbes”. A lot of the chemical reactions need improvement (e.g., superscription of charges, missing letters, spacing, wrong capitalization).

Original review round P3:

In the section “General methods of pyruvic acid production” the reference style sometimes changes from numbers in rectangular brackets [NUMBER] to author and year in round brackets (First author et al., 2000). The latter references are not found in the reference list at the end of the review, hence, are missing. Regarding citations, the authors cite their own publication, which is appropriate in the context of the review, but the cite the same paper multiple times: references [47], [48], and [62] are identical. Also [24] and [25] are identical. The authors have to correct this. The section on the chemical methods does not include yields of pyruvate, which makes it difficult to put it into perspective to the biotechnological methods. Essential experimental details should be added wherever suitable for clarity and understanding.

Response P3 by the authors:

Thank you very much for your reminder, I have corrected the problem of using different serial numbers for the same document. For citing my own literature, this is entirely out of the needs of this literature and not intentionally done. I hope the reviewers can understand it. I have also made changes to the changes in the reference format you mentioned.

Review 2 to response P3:

Duplicates of references have been removed. However, references used in the text in different style – For example: (Gennis and Hager, 1976) in the PoxB section or (Clark, 1989) in the LDH section – have not been corrected. Importantly, these references are not part of the list of references in the end, hence, citations are insufficient.

Original review round P4:

Further improvements should aim at but are not limited to:

The font changes in different sections, for example, “Biochemical pathways indirectly impacting pyruvate formation”, the first paragraph in “Biotechnological methods”, and “Production of pyruvate by recombinant E. coli”. Is this due compiling the manuscript from multiple documents?

The authors are asked to check the whole document for typos, spacing, paragraphs, general formatting (e.g., gene names should be italic and not capitalized; microbial strains like E. coli should be italic and capitalized), consistent use of units, introduction and subsequent use of abbreviations etc.

Response P4 by the authors:

Thank you very much, all the formatting issues such as paragraphs, fonts, gene abbreviations italics, and microorganism nouns italics have been revised. I'm very sorry for this relatively low-level question, thank you very much for pointing it out.

Review 2 to response P4:

I am sorry but I actually found a lot of insufficient/inconsistent formatting throughout the manuscript, typos etc. including the reference section. Some of the mistakes are even new and have not been present in the previous version of the article. The revision is insufficient.

Original review round P5:

Correct referencing has been addressed earlier and has to be revised. The reference in the first entry of Table 2, for example, is certainly not [53].

Response P5 by the authors:

I have reworked the references to correspond to the original.

Review 2 to response P5:

Entry 1 in Table 2 features E. coli TC44. It is still referenced as in the previous version with [53]: Hollmann and Deckwer (2004): Pyruvate formation and suppression in recombinant Bacillus megaterium cultivation, Journal of Biotechnology, Volume 111, Issue 1, 89–96 (DOI: 10.1016/j.jbiotec.2004.03.006) – This is clearly the wrong reference and has not been revised obviously in the resubmitted manuscript.

Original review round P6:

The submitted review by W. Yuan et al. needs in depth revision and careful contextualization of the field. Further, it is hard to grasp future trends from the final summary, which recaps mostly established concepts. All in all, publication should only be considered after extensive revision.

Response P6:

I'm very sorry for the absence of this part of a review that should have added my own views on the future at the end. I have combined the current production of pyruvate and my own views and added my own future prospects for the use of metabolic engineering and gene editing tools to transform microorganisms to produce pyruvate at the end of the article.

Review 2 to response P6:

Thank you for the addition. Nonetheless, in my opinion, the overall summary still lacks a critical assessment of the field in recent years.

Furthermore, the section “Chemical methods” does not contain a single reference. Importantly, the quality/resolution of Figure 1 has to be improved. If it is reproduced from an original or directly taken, this needs to be referenced clearly.

Author Response

Original review round P1:

Wei Yuan and co-workers composed a review addressing the production of pyruvate in different microbial hosts. Although the authors expanded the amount of relevant information beyond a previously published review on “Biotechnological routes to pyruvate production” by Ping Xu et al. (2008; DOI: 10.1263/jbb.105.169), the submitted manuscript could be improved on various levels.

The abstract already contains repeated – in other words: redundant or only slightly modified – information, for example, the higher costs of chemical pyruvate synthesis and the advantages of fermentative processes employing (engineered) microorganisms. Repeated/redundant information is flaw present throughout all sections of review and should be critically improved. Further, the abstract specifically highlights yeast, suggesting that the review will focus on yeasts, which is actually not the case. Re-phrasing is strictly recommended.

Response P1 by the authors:

Many thanks for your constructive suggestions. After consideration, I agree with you. According to your opinion, I have cut out the relatively repeated parts without affecting the reader's reading. The pyruvate-producing microorganisms described in this review are not only yeast but also microorganisms such as Escherichia coli.

Review 2 to response P1:

The abstract has been slightly modified. Whereas the addition of E. coli is suitable, why not referring to a more general phrasing like “microorganisms such as different yeasts or E. coli” since the review also includes studies with Bacillus strains, for example? What has not been addressed is the inclusion of the previous review by Ping Xu et al. (2008; DOI: 10.1263/jbb.105.169). I think, this would be valid since a good amount of information in Table 1 and Table 2 has already been reviewed before. Further, the titles of the two tables “Recombinant E. coli strains used for pyruvate production” and “Pyruvate production from recombinant microbes” should be improved; Table 2 also features E. coli besides other organisms and both titles claim to highlight recombinant microorganisms.

Response Review 2:

Thank you very much for your valuable suggestions. I have modified the titles of these two tables

Original review round P2:

The two sections after the introduction addresses biochemical pathways directly/indirectly influencing pyruvate yields in vivo. To improve this section beyond just a list of biochemical (textbook) reactions, the authors are asked to implement and describe how scientists have used metabolic engineering to reroute fluxes targeting these reactions and reference appropriate examples. This section could also be improved by balancing the amount of information. Some subsections brim over with details (e.g., size of protein subunits), whereas other sections lack information, for example, the description of uncommon terms (e.g., Bi-Uni-Uni-Bi ping-pong reaction).

Response P2 by the authors:

Thank you very much for your reminder, I have corrected the unreasonable technical terms in the article. I also add some examples of how scientists balance metabolic fluxes through metabolic engineering. For example, heat treatment of M. robertsii increases pyruvate kinase activity and thereby increases pyruvate production. Likewise, downregulation of genes such as MAA_02871 and MAA_08787 enhanced pyruvate production. Mutation of phosphoenolpyruvate carboxylase (ppc), pyruvate dehydrogenase (aceF), and alcohol dehydrogenase (adhE) in E. coli enhanced pyruvate production. In addition, thiamine or lipoic acid auxotrophs have been found to accumulate pyruvate. Auxotrophs grown in lipoic acid- or thiamine-limited media reduced the specific activity of the pyruvate dehydrogenase complex (PDHC), thereby, enhancing pyruvate accumulation.

Review 2 to response P2:

Thank you for partly editing the information content and the addition of this paragraph to the section “Biochemical pathways directly impacting pyruvate”. However, my main criticism was – and still is – that the chapter “Biochemical pathways involved in pyruvic acid” – I would change it to “Biochemical pathways involving pyruvate”. – including a list of reactions are not put into context critically enough with the next chapter “Production of pyruvate from recombinant microbes”. – I would change it to “Production of pyruvate by recombinant microbes”. A lot of the chemical reactions need improvement (e.g., superscription of charges, missing letters, spacing, wrong capitalization).

 Response Review 2:

Thank you very much for your question. I have solved this problem

Original review round P3:

In the section “General methods of pyruvic acid production” the reference style sometimes changes from numbers in rectangular brackets [NUMBER] to author and year in round brackets (First author et al., 2000). The latter references are not found in the reference list at the end of the review, hence, are missing. Regarding citations, the authors cite their own publication, which is appropriate in the context of the review, but the cite the same paper multiple times: references [47], [48], and [62] are identical. Also [24] and [25] are identical. The authors have to correct this. The section on the chemical methods does not include yields of pyruvate, which makes it difficult to put it into perspective to the biotechnological methods. Essential experimental details should be added wherever suitable for clarity and understanding.

Response P3 by the authors:

Thank you very much for your reminder, I have corrected the problem of using different serial numbers for the same document. For citing my own literature, this is entirely out of the needs of this literature and not intentionally done. I hope the reviewers can understand it. I have also made changes to the changes in the reference format you mentioned.

Review 2 to response P3:

Duplicates of references have been removed. However, references used in the text in different style – For example: (Gennis and Hager, 1976) in the PoxB section or (Clark, 1989) in the LDH section – have not been corrected. Importantly, these references are not part of the list of references in the end, hence, citations are insufficient.

  Response Review 2:

Thank you very much for your questions. I have unified the format of references

Original review round P4:

Further improvements should aim at but are not limited to:

The font changes in different sections, for example, “Biochemical pathways indirectly impacting pyruvate formation”, the first paragraph in “Biotechnological methods”, and “Production of pyruvate by recombinant E. coli”. Is this due compiling the manuscript from multiple documents?

The authors are asked to check the whole document for typos, spacing, paragraphs, general formatting (e.g., gene names should be italic and not capitalized; microbial strains like E. coli should be italic and capitalized), consistent use of units, introduction and subsequent use of abbreviations etc.

Response P4 by the authors:

Thank you very much, all the formatting issues such as paragraphs, fonts, gene abbreviations italics, and microorganism nouns italics have been revised. I'm very sorry for this relatively low-level question, thank you very much for pointing it out.

Review 2 to response P4:

I am sorry but I actually found a lot of insufficient/inconsistent formatting throughout the manuscript, typos etc. including the reference section. Some of the mistakes are even new and have not been present in the previous version of the article. The revision is insufficient.

Response Review 2:

I'm sorry that I haven't corrected all the grammatical and structural errors in the article before. I have realized these errors and corrected them this time

Original review round P5:

Correct referencing has been addressed earlier and has to be revised. The reference in the first entry of Table 2, for example, is certainly not [53].

Response P5 by the authors:

I have reworked the references to correspond to the original.

Review 2 to response P5:

Entry 1 in Table 2 features E. coli TC44. It is still referenced as in the previous version with [53]: Hollmann and Deckwer (2004): Pyruvate formation and suppression in recombinant Bacillus megaterium cultivation, Journal of Biotechnology, Volume 111, Issue 1, 89–96 (DOI: 10.1016/j.jbiotec.2004.03.006) – This is clearly the wrong reference and has not been revised obviously in the resubmitted manuscript.

Response Review 2:

I'm sorry I didn't solve this problem before, but this time I have solved it

Original review round P6:

The submitted review by W. Yuan et al. needs in depth revision and careful contextualization of the field. Further, it is hard to grasp future trends from the final summary, which recaps mostly established concepts. All in all, publication should only be considered after extensive revision.

Response P6:

I'm very sorry for the absence of this part of a review that should have added my own views on the future at the end. I have combined the current production of pyruvate and my own views and added my own future prospects for the use of metabolic engineering and gene editing tools to transform microorganisms to produce pyruvate at the end of the article.

Review 2 to response P6:

Thank you for the addition. Nonetheless, in my opinion, the overall summary still lacks a critical assessment of the field in recent years.

Response Review 2:

Thank you very much for your correction. Recently, through reading a lot of literature and combining my understanding of the bioengineering field, I put forward my own ideas on the future development of the pyruvic acid industry

Reviewer 2 Report

The corrected manuscript can be published without any corrections.

Author Response

Thank you very much for your question. I have solved the syntax and structure errors in the article
